# Environmental Regulation, Government Subsidies, and Green Technology Innovation—A Provincial Panel Data Analysis from China

**DOI:** 10.3390/ijerph182211991

**Published:** 2021-11-15

**Authors:** Pei Wang, Cong Dong, Nan Chen, Ming Qi, Shucheng Yang, Amuji Bridget Nnenna, Wenxin Li

**Affiliations:** 1School of Business Administration, China University of Petroleum-Beijing, Beijing 102249, China; wangpei@cup.edu.cn (P.W.); chennan3682@163.com (N.C.); qiming@cup.edu.cn (M.Q.); 2021211455@student.cup.edu.cn (S.Y.); 2017030046@student.cup.edu.cn (A.B.N.); 2021211454@student.cup.edu.cn (W.L.); 2School of International Trade and Economics, University of International Business and Economics, Beijing 100029, China

**Keywords:** environmental regulation, government subsidies, green technology innovation, China

## Abstract

Economic development in the “new era” will require green innovation. To encourage the growth of green technology innovation, it has become fashionable to strengthen environmental regulation. However, the impact of environmental regulation on green technology innovation, as well as the role of government subsidies, needs to be examined. Utilizing fixed-effect models and 2SLS models to explore the impact of environmental regulation on green technology innovation in China from 2003 to 2017, this research sought to examine whether environmental regulations impact green technology innovation, as well as the role of government subsidies in the above-mentioned influence path. The findings support the Porter Hypothesis by demonstrating an inverted “U” relationship between environmental regulation and green technology innovation. The impact of environmental regulation on green technology innovation varies by region. To be specific, there is an inverted “U” relationship between environmental regulation and green technology innovation in China’s central and central coast regions. In comparison, the north area, southern coast, and southwest region exhibit a “U” relationship between the two. The relationship is not significant in the Beijing-Tianjin region. Additionally, government subsidies act as an intermediate in this process, positively influencing firms to pursue green technology innovation during the earliest stages of environmental regulation strengthening. However, government subsidies above a certain level are unproductive and should be used appropriately and phased off in due course.

## 1. Introduction

After 40 years of reform and opening-up, China’s economy has entered a “new era” of pursuing high-quality growth. China’s GDP growth has been slowing down in recent years. Since 2011, GDP growth began to slow, and the growth was only 5.95% in 2019. Meanwhile, government spending on scientific and technology innovation, ecology, and education increased year by year. For example, from 2016 to 2018, the total amount of government spending on ecological and environmental protection was CNY 2.451 trillion, with an average annual growth rate of 14.8%. From 2012 to 2021, CNY 7.07 trillion was spent on science and technology in the national general public budget. This means that the economic development pattern has shifted from extensive growth of scale and speed to intensive growth of quality and efficiency. The driving force of economic growth has also changed from the factor-driven, investment-driven pattern to the innovation-driven pattern [1]. A high-quality development goal of “innovation, coordination, sustainability, openness, and sharing” was established by the Chinese government in 2017. It calls for the establishment of an economic system that promotes green, low-carbon, and circular development, as well as economic growth that is commensurate with the country’s resource carrying capacity [2]. From 1 January 2018, China’s first “green tax system,” the Environmental Protection Tax Law, came into effect, reflecting the strengthening trend of environmental regulation. In the Government Work Report in 2021, the State Council set the goal of achieving peak carbon dioxide emissions and carbon neutrality [3]. Under such circumstances, enterprises must take measures to realize coordinated and sustainable economic and ecological development [4].

Based on the Porter Hypothesis, environmental regulation will force enterprises to carry out technology innovation, which will help enterprises to obtain innovation compensation and a first-mover advantage [5]. In the current context of strengthened environmental regulations, will enterprises carry out green technology innovation? If so, what is the relationship between the two? Considering the uneven regional development in China, does the impact of environmental regulation on green technology innovation vary from region to region? 

Additionally, while firms’ technology innovation efforts have positive externalities such as increased resource efficiency and economic development (Long and Summers, 1991), there are also risks such as long investment return cycle and competitors’ imitation [6,7]. Government subsidies alleviate firms’ need to reduce pollution costs during the process of green technology innovation while also conveying information to the outside world via the “certification effect” of sponsored enterprises, thereby attracting external investment [8,9,10]. If environmental regulation impacts technology innovation, what role will government subsidies play between them? 

The answers to these questions will enrich the empirical research of the policy evaluation of environmental regulation. This paper also provides constructive suggestions to promote the further improvement and effective implementation of China’s environmental regulation policies, and to prompt enterprises to seize the opportunity of environmental tax reform to achieve high-quality development goals.

This study, therefore, contributes to the existing literature in three aspects. First, this paper integrated environmental regulation, government subsidies, and green technology innovation into the same research framework. Based on the Porter Hypothesis, the research demonstrated that government subsidies are not perfect auxiliary tools and should be withdrawn at the proper time. Second, this paper found that the influence of environmental regulation on green technology innovation is heterogeneous among the regions. Specifically, China’s central and central coast region has an inverted “U” relationship between environmental regulation and green technology innovation, whereas in the north region, southern coast, and southwest region, there is a “U” relationship between the two. In the Beijing-Tianjin area, the relationship is not significant. Third, based on the regional differences in environmental regulation affecting green technology innovation, and the limitations of government subsidies incentives, this paper provides strong support for the government to implement local policies and reasonable response subsidies and makes reliable suggestions for enterprises to respond positively to environmental regulation.

The rest of this paper is organized as follows. Section 2 reviews the relevant literature. Section 3 puts forward the hypothesis. Section 4 details the model and data. Section 5 presents the estimation techniques and empirical results. Section 6 presents conclusions and policy implications.

## 2. Literature Review

Environmental regulation is an effective way to solve the negative externalities of enterprise environmental pollution. It can be divided into three types: command control type, market incentive type, and voluntary type [11], and gradual changes from command type to market and voluntary type [12,13]. The impact of different types of environmental regulation on green technology innovation is heterogeneous. Compared with command-controlled environmental regulation, market-based environmental regulatory tools provide firms with higher incentives for green technology innovation [12].

There are mainly two views on the relationship between environmental regulation and green technology innovation. The traditional view is that environmental regulation will increase the cost of production for polluters, thereby inhibiting green technology innovation [14,15,16,17]. However, this traditional paradigm has been challenged by many scholars. The Porter Hypothesis, which is the most well-known of such theories, asserts that environmental regulations may compel firms to innovate in order to offset the costs of pollution control [5,18]. Scholars have classified the Porter Hypothesis into two categories: “strong” and “weak,” to perform further research into it. According to the “weak” Porter Hypothesis, correctly conceived environmental regulation can drive innovation [19,20,21,22,23]. In contrast, the “strong” Porter Hypothesis asserts that this innovation frequently outweighs any additional regulatory costs—in other words, environmental regulation frequently contributes to a rise in firm competitiveness [15,19]. Meanwhile, because the impact of environmental regulation on green technology innovation is dynamic and is the result of the combined effect of extrusion and stimulation [10], some researchers have discovered nonlinear relationships, such as the “U” type [3,24,25] and the inverted “U” type [26].

The new growth theory emphasizes that knowledge and human capital are “engines of growth” and holds that it is the pursuit of monopoly profits and the temporary nature of monopoly profits that keeps innovation going, thus allowing the economy to enter a sustained long-term growth [27,28,29]. Despite the obvious positive externalities associated with technology innovation activities [30,31], enterprises’ enthusiasm for R&D investment is diminished due to risks such as a lengthy investment return cycle, competitor imitation, and a high degree of uncertainty [32,33]. The government needs to take adequate measures to correct market failure, regain Pareto optimality, and internalize externalities [32,34]. Governments in various countries generally encourage private enterprises to increase research and development investment with innovative subsidy policies [35]. Direct government subsidy is one of the typical subsidy tools, which is direct ex-ante support for long-term innovation projects, namely innovation input subsidy [24]. Innovation input subsidies are a kind of supportive government subsidy that aims to promote the innovation behavior of enterprises and improve the initiative of innovation [9,10,36]. In terms of the combined effect of government subsidies and environmental regulation on green technology innovation, some scholars believe that government subsidies can mitigate the negative impact of environmental regulation [8,37,38], while others argue that government subsidies are not always effective [39]. 

Based on the existing research, this paper concludes the following: First, the Porter Hypothesis is the mainstream theory to study the relationship between environmental regulation and green technology innovation, but it is controversial in terms of establishment conditions, content explanation, and scope of adaptation. Second, most of the current studies have used emission charges data or self-designed environmental regulation indicators, but few of them could fully reflect the degree of environmental regulation. Third, many scholars have defined “green technology innovation” with innovation output or input, but few have taken “green” into consideration. Fourth, government subsidies are widely regarded as an important tool to motivate technology innovation, but few researchers have explored its role in the path of the environmental regulation affecting green technology innovation.

## 3. Research Hypothesis

Environmental regulation has two effects on enterprise-level green technology innovation. The first is the incentive effect of “innovation compensation” [5,40], which refers to the fact that environmental regulation compels businesses to lead in green technology innovation [41,42]. It enables the firms to maintain a strong competitive edge in terms of consumer loyalty, market share, and so on. The second is the crowding-out effect described by the “compliance cost theory” [14,16]. This means that the adoption of advanced equipment and technologies to meet energy-saving and emission-reduction targets will crowd out green technology innovation and R&D investment [15,43]. The combined effect of extrusion and stimulation on green technology innovation is demonstrated in Figure 1. The reduced cost of pollution management is within the enterprise’s budget during the early stages of environmental regulation. Simultaneously, the state emphasizes ecological protection, raising enterprise environmental awareness; and, due to “innovation compensation,” enterprises implement green technology innovations to offset costs and gain a first-mover advantage, implying that environmental regulation is beneficial. However, when environmental regulations become harsher, the stress associated with pollution control costs grows, and the innovation input exacerbates the cost. Meanwhile, businesses may face a situation where their innovation edge is dwindling, and competition is growing [44]. In addition, as a result of the uncertainty surrounding innovation output and other features, businesses have reduced their drive to innovate, resulting in a slowdown in innovation compensation income growth. Thus, if the expense of pollution control exceeds the revenue generated by innovation compensation, further tightening environmental regulation is counterproductive [26,45,46]. In other words, the effect of crowding out is more significant than the effect of incentives. The entire process exemplifies the relationship between environmental regulation and enterprise-led green technology innovation [47,48]. This paper presents hypothesis 1 based on the analysis presented above.

**Hypothesis** **1** **(H1).**
*There is an inverted “U” relationship between environmental regulation and green technology innovation.*


Many factors influence the impact of environmental regulations on green technology innovation at the regional level, and these aspects are multifaceted. In terms of environmental conditions, environmental quality and regulatory rigor differ by area [49]. Technically, there are also regional variances in industrial efficiency and pollution control approaches. Additionally, it cannot be disregarded that regional differences in economic development, industrial structure, energy consumption structure, and human capital exist. The disparate growth of different regions will affect the function of environmental regulation, which will affect green technology innovation in various locations. As a result, this article proposes the second hypothesis (H2).

**Hypothesis** **2** **(H2).***The impact of environmental regulations on green technology innovation varies by region*.

Although environmental regulation, according to the Porter Hypothesis, will boost enterprise green technology innovation, it cannot be overlooked that the risks associated with technology innovation activities limit entrepreneurs’ enthusiasm for technology research and development [32]. At this point, government intervention is critical in compensating for the expense of regulation to stimulate technological innovation [34]. Increased government subsidies will foster green technology innovation under stronger environmental regulation [10]. Additionally, according to the “threat-rigidity” theory, firms’ innovation activities tend to impose a financial burden on them, prompting them to pursue conservative R&D methods to avoid additional risks [6,7,50]. Government direct subsidies lessen the capital constraint on innovative firms and boost their risk-bearing capacity [37], and have an immediate and evident incentive effect on long-term innovation projects [24]. Government R&D funding has a significant impact on enterprise green technology innovation [8,51]. Effective innovation input subsidies can assist businesses in covering expenses, enhancing their ability to manage risks, and then pursuing a first-mover advantage in the initial stage, hence increasing innovation output [10]. Based on the foregoing analysis, this paper proposes hypothesis 3.

**Hypothesis** **3** **(H3).***Government subsidies have a mediating effect on the impact of environmental regulation and green technology innovation*.

The relationship among environmental regulation, government subsidies, and green technology innovation is shown in Figure 2.

## 4. Model Settings and Data Description

### 4.1. Model Settings

The purpose of this article was to examine the relationship between environmental regulation and green technology innovation, or, more precisely, to determine whether and how environmental regulation influences regional green technology innovation. As the relationship between the two is an inverted “U” in Hypothesis 1, the model includes the square of en_ru. The regression model (model 1), as described by Chen et al. (2019) [10], is as follows:(1)$$tec\_ino_{it}=\beta_{0}+\beta_{1}en\_ru_{it}+\beta_{2}en\_ru_{it}^{2}+\sum^{\;}\delta_{i}X_{it}+\alpha_{i}+\varphi_{i}+\mu_{it}.\;$$
where subscript i indicates province and t denotes time; $\beta_{2},\beta_{1}$, and $\delta_{i}$. are the parameters to be estimated in the model; $tec\_ino_{it}$ denotes green technology innovation; en_ru represents environmental regulation. $\sum^{\;}X_{it}$ refers to all control variables including the size of regional industry, level of regional economic development, human capital, and foreign direct investment. $\alpha_{i}$ describes the individual fixed effect; $\varphi_{i}$ describes the time fixed effect; $\beta_{0}$ and $\mu_{i}$ denote the constant term and random error term, respectively. Based on the inverted “U” relationship between environmental regulation and green technology innovation, the coefficient of $en\_ru^{2}$. is expected to be negative.

In order to examine the regional heterogeneity of environmental regulations’ impact on green technology innovation, model 1 is regressed in groups.

This article introduces government subsidies as a mediator and adapts sequential regression models (models 2 and 3) to examine whether government subsidies have an intermediary influence on the impact of environmental regulation on green technology innovation. Thus, according to Kashdan and Breen (2007) [52], the following models are proposed:(2)$$gov\_sub_{it}=\beta_{0}+\beta_{1}en\_ru_{it}+\beta_{2}en\_ru_{it}^{2}+\sum^{\;}\delta_{i}X_{it}+\alpha_{i}+\varphi_{i}+\mu_{it}$$
(3)$$tec\_ino_{it}=\beta_{0}+\beta_{1}en\_ru_{it}+\beta_{2}en\_ru_{it}^{2}+\beta_{3}gov\_sub_{it}+\beta_{4}en\_ru_{it}*gov\_sub_{it}+\sum^{\;}\delta_{i}X_{it}+\alpha_{i}+\varphi_{i}+\mu_{it}$$
where gov_sub. denotes government subsidies. Due to the government subsidies that encourage enterprises to pursue a first-mover advantage in the initial stage, the coefficient of the square of en_ru is expected to be negative, while the coefficient of interaction item is expected to be positive in model 3.

### 4.2. Data

To examine the impacts of the environmental regulation on green technology innovation and analyze the role of government subsidies, a balanced panel dataset of 25 provinces and 4 cities for 2003–2017 was utilized in this study. In addition, due to the uneven regional development in China, the 29 provinces (cities) were divided into eight regions, i.e., Beijing–Tianjin, North, Central, Central Coast, South Coast, Northwest, Southwest, and Northeast [53,54]. The detail of provinces (cities) in the eight regions are shown in Table 1.

To avoid the influence of missing values, 435 samples were obtained by excluding Tibet, Qinghai, Hong Kong, Macao, and Taiwan. All the data used in this paper were taken from China Statistical Yearbook, China Science and Technology Statistical Yearbook, China Environment Statistical Yearbook, China Energy Statistical Yearbook, China Labor Statistical Yearbook, and China Industrial Statistical Yearbook.

The following is the description of the variables in this paper.

#### 4.2.1. Dependent Variable

Green technology innovation. Green technology innovation is typically described in terms of innovation output indicators (such as the number of patents or the income generated by new products) or innovation input indicators (such as research and development input and investment in technological transformation funds). However, relying just on innovation output or input indicators fails to reflect “green.” On the other hand, output indicators can more adequately describe the transition from technology to innovation. Guo et al. (2018) [25] expanded on Awasthi et al. (2010)’s [55] and Yuan et al. (2010)’s [56], definitions of green technology innovation by stating that green technology innovation may be classified into product innovation and technological process innovation [57,58,59]. The term “new product” refers to an altogether novel concept developed employing a novel technical principle or design approach in this article. On the other hand, it includes advancements in design, material, and technology. As a result, the revenue earned by novel products accurately reflects the innovation in products and technological processes. In comparison to traditional innovation, green technology innovation places a higher premium on energy conservation and consumption reduction throughout the product’s life cycle and is committed to maximizing output per unit of energy consumed [60,61]. Thus, the output per unit of energy usage is frequently employed in research to quantify green technology advancement [25]. As a result, to capture the “green” aspect of innovation output, this research uses the logarithm of innovation output per unit of energy consumption (new product sales revenue/total energy consumption) as the measurement index. The ratio means that the energy consumption, as input, drives the output of new product sales revenue, and can positively reflect green technology innovation. At the same time, the results in Figure 3, Figure 4 and Figure 5 indicate that the index is affected by neither the price increase nor the reduction in energy consumption. The higher the index value, the greater the capacity for green technology innovation.

#### 4.2.2. Independent Variable

Environmental regulation. There are numerous types of environmental regulation indices available at the moment. To define environmental regulation, some researchers began with the source of regulation, such as the change in the pollution emission baseline [62] and the quantity of environmental regulation or law [58]. However, as is frequently found in developing economies, the legislation does not always implement efficient environmental regulation. In other words, there is frequently a phenomenon known as “paper law” in those countries, showing a significant disconnect between legislation and enforcement [63]. As a result, some scholars have turned to indicators that can accurately reflect the outcomes of environmental regulatory implementation, such as the cost of pollution control paid to specific trash [64] and the quantity of pollution control investment made by the organization [65]. In addition, Hernandez-Sancho et al. (2000) [66] argued that the stronger the environmental regulation, the less pollution the enterprise will emit. Therefore, the pollutant emissions per unit of enterprise output [67,68] and the integrated pollutant removal efficiency index [46,63] become the proxy indicators for environmental regulation. Consequently, in terms of the selection of environmental regulation variables, this paper used the method of Yuan and Xie (2014) [69] and Chen et al. (2019) [10] to construct a comprehensive measurement system by using the ratio data of wastewater discharge, solid waste discharge, sulfur dioxide (SO_2_), and soot emission to regional GDP. In other words, weights are assigned to different pollutants according to the formula, $w_{it}=\left(E_{ijk}/\sum^{\;}E_{ijk}\right)/\left(G_{ij}/\sum^{\;}G_{ij}\right)$. $E_{ijk}$ represents the corresponding emissions of a certain pollutant in different regions in different years, and $G_{ij}$ represents the GDP of a certain region in corresponding years; then, the measures are normalized and reciprocal. The larger the index value, the stronger the degree of environmental regulation.

#### 4.2.3. Intermediate Variable

Government subsidies. Most scholars have defined this indicator in terms of the amount of government assistance received by businesses during the present era. As a measurement index, the logarithm of government funds in internal R&D expenditures of industrial businesses is utilized. The expenditure of enterprise R&D funds is classified as internal and external. Internal expenditures include all costs incurred by the firm to conduct R&D activities. The term “external expenditure” refers to the amount paid to the enterprise for external research and development operations. Internal R&D expenditures are more directly tied to corporate development. Government grants in this category can indicate the government’s support for enterprise investment in green technology innovation: the greater the index value, the more robust the government’s support for corporate research and development.

#### 4.2.4. Control Variables

Size of regional industry. The regional sewage discharge scale and governance capacity are closely related to the regional industrial scale, so we take it as one of the control variables. According to the National Bureau of Statistics of China, “above-scale” enterprise refers to an industrial legal entity with an annual income of CNY 20 million or more from its main business, and the reason why we use this indicator to measure regional industry size is that the output value of “above-scale” enterprises accounts for more than 90% of the total output value of provincial industrial enterprises, which plays a pivotal role in the provincial economy. Meanwhile, this part of the enterprise information data is also the focus of national statistics. Therefore, the number of above-scale industrial enterprises in each region after logarithm transformation is used to measure the regional industrial scale.

Level of regional economic development. The economic development level of each province is an important influencing factor of its regional green technology innovation ability. This paper uses the per capita GDP of 29 provinces to control the economic development difference of each province. The year 2000 is used as the base period, and the per capita gross domestic product over the years is deflated in order to counteract the effects of inflation.

Human capital. The human capital of each province has a significant difference. It is an important factor that affects the output of green technology innovation. This paper selects the number of years of education per capita as the proxy variable of human capital. According to the proportion of people in different stages of education, the corresponding number of years of education is empowered to obtain the number of years of education per capita.

Regional population. The population of each province is closely related to its economic and technological development. This indicator is the logarithm of the total population of a region.

Foreign direct investment. The more robust the green technology innovation, the more capable firms are of dealing with environmental regulations, and thus the more competitive enterprises are in attracting foreign investment [49,70]. The impact of foreign investment on firms’ green technology innovation should be controlled. As a result, the logarithm of foreign direct investment was utilized as the measuring index in this work.

Detailed descriptions of this study’s variables are also presented in Table 2.

### 4.3. Descriptive Statistics

The descriptive statistics for variables are shown in Table 3. The average value of green technological innovation (tec_ino) is found to be 7.016935, with a standard deviation of 1.272259. The general level of green technology innovation among industrial businesses greater than the scale of each province fluctuates slightly. Environmental regulation (en_ru) has a low standard deviation and an average, indicating that the general difference in environmental regulation is minimal. Government subsidies (gov-sub) have an average value of 12.36613 and a low standard deviation. Other control variables, such as regional economic development (per-GDP), the number of regional industries (size), and foreign direct investment (FDI), are relatively stable in aggregate (Table 3).

To examine the influence of environmental regulation on green technology innovation across regions, Figure 6 compares the mean of green technology innovation and environmental regulation in various regions. As illustrated in Figure 6b, environmental regulation is more severe in Beijing and Tianjin, the central coast, and the southern coast than in other regions, although the distinction between environmental regulation in other regions is not entirely obvious. Similarly, as illustrated in Figure 6a, Beijing and Tianjin, the central coast, and the southern coast all have a relatively high level of green technology innovation, followed by the northern and central regions. In contrast, the northeast and southwest regions outperform the northwest regions, basically consistent with the economic development. The level of green technology innovation varies significantly between regions when compared to environmental regulation. We can see from these figures that environmental regulation is also stricter in regions with a high level of green technology innovation. However, the relationship between environmental regulation and green technology innovation in various regions requires further verification.

## 5. Empirical Results and Analysis

### 5.1. Panel Unit Root Test and Cointegration Test

To avoid pseudo-regression, the LLC [71] (Levin et al., 2002) and IPS [72] tests were utilized in this research to examine whether the variables have a unit root. The result indicates that all variables are integrated on the order of zero, i.e., I (0), except for en_ru, which is integrated on the order of one, i.e., I (1). The panel data analyzed in this article are stable. Additionally, the cointegration test proves the existence of a cointegration relationship in long-run equilibrium by the use of the tests [73,74,75].

### 5.2. Hausman Test

On the one hand, the data used in this article were provincial data from 2003 to 2017, which are relatively complete. Moreover, due to the distinct geopolitics of each province, some factors may be omitted that do not change over time. As a result, the fixed-effect model is appropriate for the data analysis in this work and also contributes to the consistency of estimation. Additionally, the F test *p* value is 0.0000, indicating that the Hausman test results support the fixed-effect model’s suitability for the research [9,10].

### 5.3. Results

#### 5.3.1. The Overall Impact of Environmental Regulation on Green Technology Innovation

As illustrated in Table 4, panel regression was used to examine the impact of environmental regulation on green technology innovation. Increased regional green technology innovation may result in reduced pollutant emissions and a decrease in the level of environmental regulation. According to the study above, environmental regulation and green technology innovation are likely to have a mutual causal relationship. Endogeneity has an effect on the robustness of regression results. Our article argues that while regional environmental regulation is inextricably linked to provincial environmental regulation, regional environmental regulation cannot have a direct effect on provincial technology innovation. Thus, regional environmental regulation satisfies instrumental variable requirements. This research will further validate the fixed-effect results by utilizing regional environmental regulation as an instrumental variable. Column FE contains the regression findings for fixed effects, while column IV contains the regression results for instrumental variables.

Before examining the nonlinear relationship between environmental regulation and green technology innovation, this article examined the linear relationship between both, as illustrated in Table 4, Column 1. Both the FE and IV regression results imply that environmental regulation promotes green technology innovation, but the sustainability of this relationship remains largely unknown. Column 2 of Table 4 contains the regression results for Model 1. The coefficient of en_ru is positive, but the square term has a negative coefficient. Meanwhile, as illustrated in Table 5, the *p*-value for the Sasabuchi test of the inverse U-shape in PPC is 0.008, and the estimated extreme point (3.882) is within the 95% confidence interval. It demonstrates an inverted “U” relationship between environmental regulation and green technology innovation. Prior to the turning point, with the strengthening of environmental regulation, enterprises are forced to carry out green technology innovation and increase output. At this stage, the “innovation compensation effect” is dominant. Following the inflection point, when environmental regulation is strengthened further, the demand for businesses to sustain production increases, while the desire for innovation decreases. This tendency has demonstrated a greater degree of “innovative crowding-out effect”.

#### 5.3.2. The Regional Impact of Environmental Regulation on Green Technology Innovation

Regression grouping was performed according to region, and the findings are summarized in Table 6. The results of model 1’s regional regression reveal an inverted “U” relationship between environmental regulation and green technology innovation in the central region, a relationship that is also evident in the central coast region. Environmental regulation in these localities will incentivize firms to develop green technology innovations at an early stage. However, in the later stage, it will leave a burden to enterprises and diminish their innovative output. Compared to the center and central coast regions, the regression results indicate that the west’s environmental regulation and green technology innovation follow a “U” relationship. The reason could be that when environmental regulation is in its developing stage, cost pressures are relatively low, and government officials in the western region, for performance reasons, pay more attention to economic development indicators, implying a relative lack of environmental protection consciousness. Simultaneously, high-quality staff and other innovation resources are scarce, so they mainly rely on the extensive development pattern. As a result, when the western region of the enterprise is under the less stringent environmental regulation, they prefer to follow specific pollution spending to reduce pollutant emissions while squeezing some research and development funding rather than pursuing technology innovation. When environmental regulations become severe, the cost of pollution management will increase even more, compelling firms in the western region to achieve the goal of energy-saving and emission reduction through technology innovation. In the most economically developed (i.e., Beijing–Tianjin) and underdeveloped regions (i.e., Northwest and Northeast), the impact of environmental regulation on green technology innovation is not significant in these regions. The regression results indicate that the impact of environmental regulation on green technology innovation varies significantly by region.

#### 5.3.3. The Mediating Effect of Government Subsidies

The regression results from models 2 and 3 imply that government subsidies play an intermediary role in the impact of environmental regulation on green technology innovation. As illustrated in Table 7, environmental regulation affects green technology innovation by influencing government subsidies.

The results indicate a “U” relationship between environmental regulation and government subsidies in column 2 of Table 7. After model fitting, we found that there is a positive correlation between environmental regulation and government subsidies when the value of environmental regulation is greater than 0. Thus, based on the practically meaningful part of the fitting results, with the strengthening of environmental regulation, the government may increase investment in subsidies.

The inverted “U” relationship between environmental regulation and green technology innovation is presented in Column 1 of Table 7. Environmental regulation, before the inflection point, encourages green technology innovation through government subsidies. The strengthening of environmental regulation necessitates green technology innovation on the part of businesses. However, innovative efforts are associated with significant risks and lengthy durations, resulting in a lack of innovation motivation. As a result, government subsidies have become a critical instrument for stimulating enterprise innovation by effectively lowering the costs and risks associated with innovation. Additionally, the compensatory impact and first-mover advantage received by firms due to innovation will encourage enterprises to innovate further.

Following the inflection point, environmental regulation inhibits green technology innovation through government subsidies. Further investment in government subsidies may backfire as environmental regulation is strengthened. Excessive government subsidies would be squandered on various rent-seeking strategies [76,77]. Additionally, the ability of enterprises to transform their innovations affects the impact of government subsidies on green technology innovation [78,79]. When firms’ capacity for innovation transformation is exhausted, increasing government subsidies will have little effect on green technology innovation [80]. Indeed, businesses cannot implement green technology innovation in this period just through government subsidies.

#### 5.3.4. Further Research

According to the analysis of the regression results above, the present study indicates that environmental regulation has an inverted “U” relationship effect on green technology innovation. A robustness test was undertaken to further verify the conclusion. The logarithm of the annual number of invention patents granted in each province was used to determine the level of green technology innovation. Table 8’s first column contains the results. The robustness test result agrees with the result in Table 4. Environmental regulation’s primary term coefficient is positive, while its square term coefficient is negative, which is statistically significant at a 1% level of significance. The robustness test demonstrates that environmental regulation does have a considerable impact on green technology innovation. The inverted “U” relationship means green technology innovation can be accelerated during the earliest stages of environmental regulation strengthening.

## 6. Conclusions

The impact of environmental regulations on green technology innovation was examined in this article using data from China’s provincial panel from 2003 to 2017. Samples were grouped to investigate whether or not there is regional heterogeneity. The paper employed both a fixed effect and an intermediary effect model, and government subsidies were used as a mediating variable to examine the effect of government subsidies on green technology innovation. The following conclusions can be drawn from the findings:

Firstly, there is an inverted “U” relationship between environmental regulation and green technological innovation on a whole level. The pressure of rising pollutant discharge costs compels firms to invest in green technology innovation during the initial phase of environmental regulation strengthening. When environmental regulations are enforced further, enterprises face increased production pressure, resulting in diminished green technology innovation capacity and decreased output. 

Secondly, because economic development levels vary at the regional level, the impact of environmental regulation on green technology innovation varies among regions. The results indicate an inverted U-shaped relationship between environmental regulation and green technology innovation in the eastern and central regions; environmental regulation initially stimulates and then inhibits green technology innovation. However, the relationship is more substantial in the East region. Given that the western and northeastern regions are developing, it is apparent that environmental regulation cannot stimulate green technology innovation. Even in the west, it will inhibit innovation in green technology innovation.

Thirdly, government subsidies have a mediating effect on the impact of environmental regulation on green technology innovation. Government subsidies assist by balancing R&D risk and lowering R&D costs. Enterprises will respond positively to environmental regulation if the government subsidizes them. They may carry out green technology innovation and scale-up innovation output. However, more government subsidies are not better sometimes. When environmental regulations are strengthened to a certain extent, government subsidies cannot drive enterprises to innovate.

To sum up, the following policy recommendations are put forward for reference:

To begin, environmental control cannot be completed overnight in China’s current stage, and relevant policies should not excessively pursue high standards. Prompt feedback on the efficacy of policies may be beneficial for policy implementation and improvement. It is possible to effectively enhance firms’ environmental regulation compliance and green technology innovation by gradually improving environmental regulation.

Second, environmental regulation policies should be implemented according to local conditions and the differences in regional economic development. It is necessary to develop appropriate environmental regulation policies in the economically developed eastern region and coastal region in order to promote green technology innovation of enterprises. Meanwhile, the developing region’s environmental regulation policy should be strengthened to improve the environmental protection consciousness of the western region of enterprises and make the innovative behavior strive to infinity. As a result, these underdeveloped regions will experience rapid economic and ecological development. Simultaneously, in the face of expanding environmental regulations, businesses can be incentivized to innovate by combining the characteristics of several policies, such as government subsidies and carbon trading permits.

Third, government subsidies are an effective means of regulating the environment. This policy combination can guide firms in the early stages of dealing with environmental regulation and increasing green technology innovation. At the same time, it should be highlighted that government subsidies are not a one-size-fits-all proposition and should be phased off gradually. Government subsidies come in a variety of forms. It is prudent to use them flexibly to encourage firms to innovate green technologies and avoid the negative consequences of blindly increasing subsidies.

However, further research questions require further discussion. First, due to data collecting constraints, this paper’s sample period spanned from 2003 to 2017. If additional data had been available, this article would have produced a more significant number of robust outcomes. Second, the paper focused exclusively on geographical disparities at the macro level, omitting the impact of micro-level variances in enterprise characteristics. The environmental regulation’s influence on green technology innovation needs to be thoroughly explored in order to provide a more practical and effective reference point for government policy reform. Third, we concur with some scholars in asserting that a region’s or country’s level of green technology innovation is contingent not only on its economic development but also on the structure of the productive regional system (what type of sectors predominate in the regional industry) [81,82,83]. We did not validate this in our work, due to the difficulty in acquiring relevant data. However, a microscopic examination of the question is necessary to establish a direction for additional research.

## Figures and Tables

**Figure 1 ijerph-18-11991-f001:**
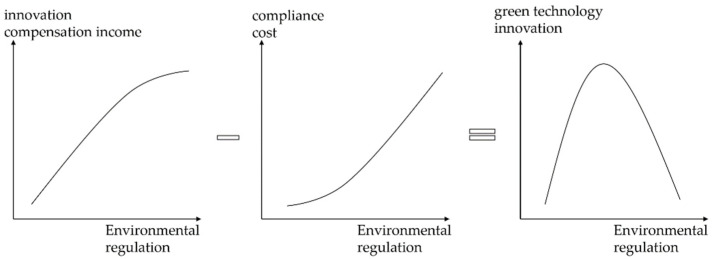
The additive combinations of latent mechanisms result in an inverted U-shaped relationship between environmental regulation and green technology innovation.

**Figure 2 ijerph-18-11991-f002:**
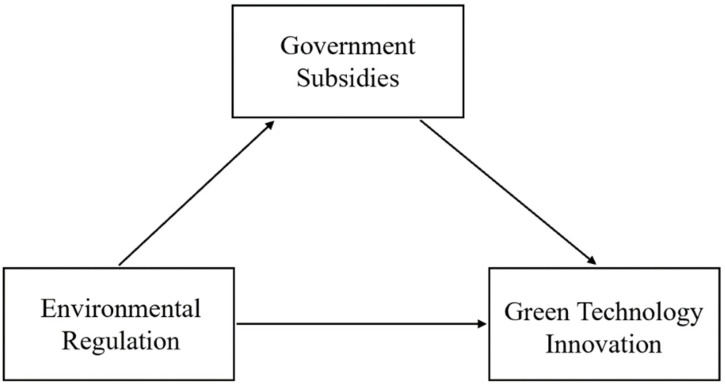
The intermediary role of government subsidies in the path of environmental regulation affecting green technology innovation.

**Figure 3 ijerph-18-11991-f003:**
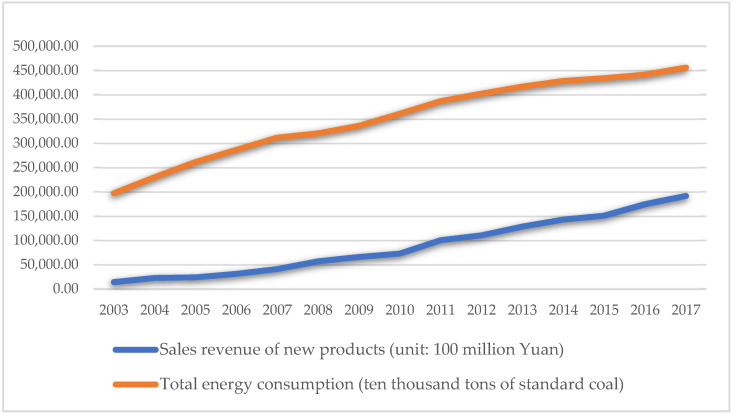
2003–2017 sales revenue of new products and total energy consumption (data source: China National Bureau of Statistics).

**Figure 4 ijerph-18-11991-f004:**
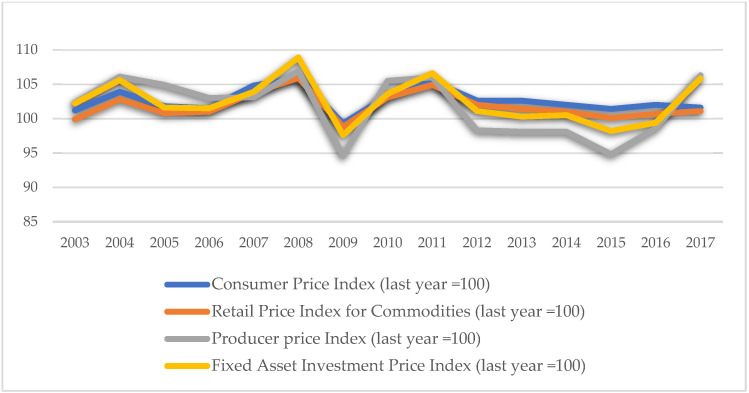
2003–2017 price index (data source: China National Bureau of Statistics).

**Figure 5 ijerph-18-11991-f005:**
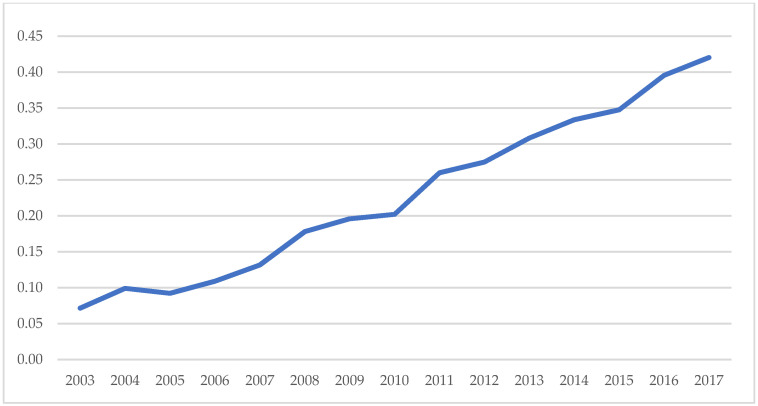
2003–2017 new product sales revenue per unit of energy consumption (data source: China National Bureau of Statistics).

**Figure 6 ijerph-18-11991-f006:**
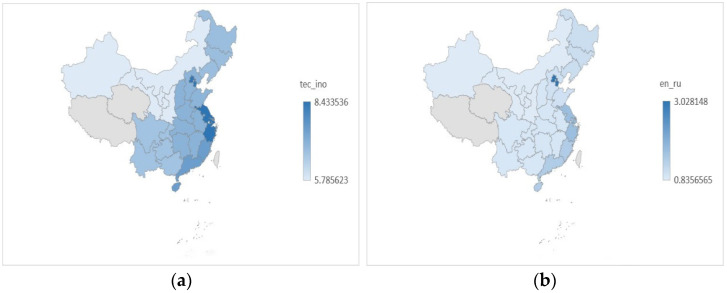
(**a**) The mean of green technology innovation in eight regions; (**b**) the mean of environmental regulation in eight regions. In these figures, the grey parts represent Qinghai and Tibet. Due to lack of data, the corresponding index values are not shown in the figure above.

**Table 1 ijerph-18-11991-t001:** The provinces (cities) in eight regions of China.

Regions	Provinces (Cities)
Beijing–Tianjin	Beijing, Tianjin
North	Hebei, Shandong
Central	Henan, Shanxi, Anhui, Hunan, Hubei, Jiangxi
Central Coast	Shanghai, Zhejiang, Jiangsu
South Coast	Guangdong, Fujian, Hainan
Northwest	Inner Mongolia, Shaanxi, Gansu, Ningxia, Qinghai, Xinjiang
Southwest	Sichuan, Chongqing, Yunnan, Guizhou, Guangxi
Northeast	Heilongjiang, Jilin, Liaoning

**Table 2 ijerph-18-11991-t002:** Description of the variables.

Variable Name	Variable Code	Metrics	Variable Property
Green technology innovation	tec_ino	new product sales revenue/total energy consumption	Dependent variable
Environmental regulation	en_ru	Calculated by comprehensive weighting method	Independent variable
Government subsidies	gov_sub	The government funds in the internal expenditures of R&D funds after logarithmic transformation	Intermediate variable
Size of regional industry	size	the number of above-scale industrial enterprises after logarithmic transformation	Control variable
Level of regional economic development	per_gdp	Per capita GDP	Control variable
Human capital	hum_cap	Number of years of education per capita	Control variable
Regional population	popu	take log of total population of a region	Control variable
Foreign direct investment	fdi	take log of foreign direct investment	Control variable

**Table 3 ijerph-18-11991-t003:** Descriptive statistics of the variables (after logarithm).

Variable	Observations	Mean	SD	Min	Max
tec_ino	435	7.016935	1.272259	3.72145	9.170255
en_ru	435	1.199214	1.097396	0.1324278	5.514703
gov-sub	435	12.36613	1.573007	8.550241	15.6511
Size	435	8.81462	1.122623	5.940171	10.98217
per-gdp	435	10.25625	0.7188959	8.677609	11.66635
hum-cap	435	9.792706	0.882625	7.77328	13.0702
popu	435	8.227627	0.6788662	6.413459	9.272752
Fdi	435	12.40906	1.65686	8.171599	14.86275

**Table 4 ijerph-18-11991-t004:** Total sample regression.

Dependent Variable: tec_ino				
	(1)		(2)	
	FE	IV	FE	IV
en_ru	0.515 ***	0.133 **	1.490 ***	2.413 ***
	(4.05)	(2.13)	(6.90)	(3.54)
en_rusq			−0.192 ***	−0.366 ***
			(−4.89)	(−3.28)
Size	−0.292 ***	−0.005	−0.279 **	−0.150 **
	(−2.78)	(−0.13)	(−2.43)	(−2.05)
per_gdp	−0.330	0.032	−0.857 ***	−0.303 *
	(−1.66)	(0.34)	(−3.94)	(−1.72)
hum_cap	−0.151	−0.142 ***	0.016	0.085
	(−1.48)	(−5.33)	(0.15)	(1.12)
Popu	−1.639	0.378	−1.835	0.488
	(−1.64)	(1.04)	(−1.67)	(0.71)
Fdi	0.129	0.056 *	0.199 **	0.220 ***
	(1.44)	(1.66)	(2.06)	(3.31)
		(−0.36)		(−0.67)
Constant	25.720 ***		29.460 ***	
	(3.19)		(3.39)	
Observations	435	435	435	435
R-squared	0.208	0.858	0.297	0.630
pro	29	29	29	29

Note: ***, **, and * denote statistical significance at 1%, 5%, and 10%, respectively, and the values in parentheses represent t-statistics.

**Table 5 ijerph-18-11991-t005:** Test of an inversely U-shaped relationship between environmental regulation and green technology innovation.

	Green Technology Innovation
Test of joint significance of PPC variables (PPC and PPC-squared) (*p*-value)	0.000
Sasabuchi-test of inverse U-shape in PPC (*p*-value)^4^	0.008
Estimated extreme point	3.882
95% confidence interval—Fieller method	[3.350, 4.976]
Test of joint significance of control variables (*p*-value)	0.000
Test of joint significance of all variables in the model	0.000

**Table 6 ijerph-18-11991-t006:** The regional grouping regression.

Dependent Variable: tec_ino								
	(1)	(2)	(3)	(4)	(5)	(6)	(7)	(8)
Region	Beijing-Tianjin	North	Central	Centra-Coast	South-Coast	Northwest	South-West	Northeast
en_ru	0.413	−0.592 ***	0.761 *	1.423 ***	−0.337	0.366	−0.424 ***	−0.522
	(1.81)	(−82.41)	(2.56)	(11.28)	(−0.62)	(0.40)	(−5.08)	(−1.54)
en_rusq	−0.043	0.022 **	−0.135 *	−0.132 **	0.147 *	−0.210	0.068 **	0.202
	(−5.46)	(16.63)	(−2.25)	(−6.76)	(3.18)	(−0.72)	(3.28)	(1.91)
Size	−0.034	−0.196 ***	−0.257 **	0.244	−0.119	−0.144 *	−0.020	0.361
	(−0.05)	(−77.99)	(−2.63)	(2.37)	(−0.92)	(−2.64)	(−0.33)	(0.49)
per_gdp	−0.986 **	0.197 **	0.842 *	0.029	−0.351	0.815 *	−0.118	−1.409
	(−18.48)	(18.54)	(2.28)	(0.16)	(−1.17)	(2.14)	(−0.65)	(−0.70)
hum_cap	0.724	0.038 **	−0.265 ***	−0.008	−0.344 **	−0.144	0.050	0.244
	(2.64)	(58.50)	(−5.14)	(−0.25)	(−5.71)	(−0.99)	(0.65)	(1.63)
Popu	−6.695	−18.817 **	3.020	1.111	−0.773	3.808	−2.151 *	10.464
	(−3.10)	(−63.64)	(1.55)	(1.07)	(−0.86)	(1.54)	(−2.42)	(0.87)
Fdi	0.322	0.128 ***	0.028	−0.245	0.557 *	−0.063	0.037	0.082
	(1.37)	(65.03)	(0.43)	(−2.01)	(3.23)	(−1.21)	(0.78)	(0.62)
Constant	56.189	174.648 ***	−21.584	0.076	15.216	−26.513	25.952 **	−70.744
	(5.80)	(64.34)	(−1.14)	(0.01)	(2.28)	(−1.33)	(3.16)	(−0.73)
Observations	30	30	90	45	45	75	75	45
R-squared	0.790	0.998	0.973	0.987	0.802	0.925	0.957	0.825
Pro	2	2	6	3	3	5	5	3

Note: ***, **, and * denote statistical significance at 1%, 5%, and 10%, respectively, and the values in parentheses represent t-statistics.

**Table 7 ijerph-18-11991-t007:** The mediating effect of government subsidies.

	(1)	(2)	(3)
Dependent Variable	tec_ino	gov_sub	tec_ino
en_ru	1.490 ***	−0.918 ***	−0.500 **
	(6.90)	(−4.32)	(−2.20)
en_rusq	−0.192 ***	0.161 ***	−0.031 *
	(−4.89)	(5.60)	(−1.79)
gov_sub			−0.112
			(−1.19)
en_ru ** gov_sub			0.056 ***
			(4.21)
size	−0.279 **	0.530 ***	−0.038
	(−2.43)	(3.52)	(−0.86)
per_gdp	−0.857 ***	−1.037 ***	0.054
	(−3.94)	(−7.16)	(0.42)
hum_cap	0.016	0.136 **	−0.090 **
	(0.15)	(2.07)	(−2.26)
popu	−1.835	−3.830 **	0.760
	(−1.67)	(−2.56)	(1.37)
fdi	0.199 **	−0.128	0.071
	(2.06)	(−1.28)	(1.51)
Constant	29.460 ***	50.765 ***	2.809
	(3.39)	(4.05)	(0.52)
Observations	435	435	435
R-squared	0.297	0.756	0.868
Number of pro	29	29	29

Note: ***, **, and * denote statistical significance at 1%, 5%, and 10%, respectively, and the values in parentheses represent t-statistics.

**Table 8 ijerph-18-11991-t008:** The robust test.

	(1)
Dependent Variable	Patent
en_ru	1.151 ***
	(6.00)
en_rusq	−0.117 ***
	(−3.85)
size	0.122 *
	(1.95)
per_gdp	0.229
	(1.51)
hum_cap	−0.076
	(−1.08)
popu	0.026
	(0.06)
fdi	−0.164 ***
	(−3.62)
Constant	5.511
	(1.30)
Observations	150
R-squared	0.652
Number of pro	29

Note: *** and * denote statistical significance at 1%, 5%, and 10%, respectively, and the values in parentheses represent t-statistics.

## Data Availability

Publicly available datasets were analyzed in this study. This data can be found here: [http://www.stats.gov.cn/tjsj/ndsj/], accessed on 13 September 2021.

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
