# Peer review of "Environmental Regulation, Government Subsidies, and Green Technology Innovation—A Provincial Panel Data Analysis from China"

_ijerph, 2021, doi:10.3390/ijerph182211991_

Round 1

Reviewer 1 Report

It is the aim of the authors, to investigate the influence of environmental regulations and environmental subsidies on technological innovations, which reduce the environmental impact of production processes.

After reading the paper I have a number of serious concerns.

-The authors do not provide a good reason why they consider different Chinese provinces as research subjects. Particularly, the authors do not provide one reason why innovative behavior of firms depends on the location of the firms. The innovative behavior depends on many factors, but that the location is important seems to be a weird assumption.

-The reference list is incomplete, partly the references are incomplete and it contains references which cannot be found, like “Li, G.X., Zhang, W., Wang, Y.J. Business SO, Jinan UO. OFDI, 2016. Environmental Regulation and Green Technology Innovation in China. Sicence and Technology Management Research.”

-It looks like that the authors do not understand the economics of externalities, particularly of positive externalities. It is a few decades ago that I read Arrow (1971), but he did not argue that there is a “loss of technology spillover”. To make that clear, all educated humans use mathematics, but we do not pay a license fee for doing it, thus we benefit from a knowledge spillover, but what is the loss from the view of the mathematicians, who have developed the laws of mathematics? Thus, I recommend that the authors read again a good textbook which contains a chapter on externalities. (Like Mankiw’s Principles of Economics or Krugman & Well’s “Economics”.). However, it is long time ago that Arrow (1971) was state of the art, a lot of fundamental research have been done in the field, particularly I refer to work of the Nobel Prize laureates Paul Romer (Romer, P. M. (1986). “Increasing Returns and Long Run Growth.” Journal of Political Economy 94: 1002-38) and Robert Lucas (Lucas, R. E. (1988). “On the Mechanics of Economic Development.” Journal of Monetary Economics 22: 3-41) as examples. Further, the authors fully ignore the literature provided by new growth theorists like Grossman, Helpman, Jones, Aghion, Howitt and so on, who investigate innovative behavior of firms and its macroeconomic consequences. Accordingly, a statement like (line 109-110), “Due to the obvious positive externalities in technological innovation activities (Long and Summers, 1991; Ma et al., 2019), the enthusiasm of enterprises for R&D investment is reduced (Nadiri, 1993; Yang et al., 2021).” is simply wrong. Correct is that subsidies can help to establish the optimal level of R & D investments. However, at the firm level these externalities are internalized, given that the outcome of R & D investments leads to patents (That is the economic reasoning why patents are important!). This aspect is fully ignored by the authors.

-Then, the authors should recognize, empirical studies make sense if they test hypotheses derived from consistent theories. An empirical result with no underlying theory makes not much sense, because then we do not know the underlying mechanisms. In this paper the authors do not provide any theory for the assumed inverted U-shaped relationship between regulation and innovative behavior of firms.

-Anyways, the idea to use “regulation” as a variable does not convince me, because it includes too many different approaches, which have partly very different consequences. For example, a command-and-control policy which requires that electricity generation does not produce nuclear waste as a side product, leads to the shut-down of nuclear power stations. In contrast, the use of an emission tax requires in a perfectly competitive market that the marginal tax rate shall be equal to the marginal damage created by the emissions. That is the only optimal tax, whatever the consequences will be for firms.

-However, what raises my main concern is the choice of indicators, because questionable indicators make empirical analysis use and senseless. The indicator for regulation is, “the comprehensive indicator is calculated by combining three pollutants (i.e., industrial sulfur dioxide (SO2) emissions, industrial wastewater discharge, and general industrial solid waste discharge) based on the entropy method and then we got the environmental regulation intensity by taking the reciprocals of those values.”. Taking a comprehensive value of pollution volumes shall measure regulation? Sorry, but I do not think that this indicator measures what shall be measured.

The indicator for innovation is defined as “new product sales” divided by total energy consumption (of what? The economy, the total production, production of the new goods?). I also argue, that this indicator does not measure “green” output of innovation. To give an indication how difficult it is to measure “green ” innovation, look at this study https://doi.org/10.1016/j.energy.2019.04.080 which states that the energy consumption or CO2 emissions of e-cars in production (13 tons CO2) exceeds the energy consumption of fossil-fueled cars in production (10 tons CO2), but if the electricity used by e-cars is generated from renewable resources, the total CO2 footprint of e-cars is much lower than the CO2 footprint of fossil-fueled cars after driving a few thousands kilometers. However, for me it is not clear what the indicator of the authors will tell us in this example. Another bias results from the fact that new products are mostly sold at higher prices than old-fashioned products, because of marketing reasons, although the quality of old and new products do not necessarily differ regarding the utility provided to consumers. Thus, the indicator is not very plausible.

-The authors use as indicator for “size of regional industry” the number of above-scale enterprises in each region. However, to measure the industry size by counting the number of firms seems to be not appropriate. Anyways, it remains unclear what “above-scale” in the context shall mean.

-For the Western regions the authors found a U-shaped relationship. That means that the hypotheses of the authors have to be rejected according to the usual standards. However, the authors try to argue in an inconsistent manner that also these results are serious and caused by the low level of economic development (this is not very logic and inconsistent what is written in the paper and it does not fit to the authors’ hypothesis.). But the results for the Western regions are in some sense nice, because the only policy recommendation is then, that the regulation measures should be as strong as possible, because then the innovative behavior will strive to infinity. Thus, these underdeveloped regions will catch-up economically and ecologically very fast with rest of China. Surprisingly, the authors recommend the opposite, that the regulation should be relaxed. That means that the authors do not take their own results seriously. That proves very well, that something is fundamentally wrong with the study.

Because of the fact, that I have strong doubts that the used indicators measure what they shall measure, I cannot take this analysis serious. Or in other words this study makes no sense and does not follow the usual scientific standards.

Minor concerns

-The introduction reads partly like a government report, but has no scientific content, example: “The pattern of economic development has shifted from extensive growth of scale and speed to intensive growth of quality and efficiency.” The sentence sounds good, but it has scientific content, because the used terms are not well-defined.

-What is an optimized “industrial structure” and optimized “energy structure”? It is new for that someone has ever developed a theory of industrial structure, how can this then be a goal of a State Council? I am aware that policy-makers everywhere on this planet tell a lot of rubbish, but this should not be overtaken in scientific papers and if it needs a corrective comment. What means “win-win situation of ecology and economy”? As long as the economy needs energy, the laws of thermodynamics inform us that we have a trade-off to solve between economy and natural environment. There is in general nothing like a win-win situation, that is an illusion created by policy-makers who do not want to accept that Nature does not make compromises. Either we accept the trade-off, or humankind as a species will die out in the following centuries.

-What are “high-quality development goals”?

-Line 147: Sinn and Hans-Werner is wrong, it is one guy and his family name is “Sinn” and the given names are “Hans-Werner”. Thus, the name is Hans-Werner Sinn.

-What have the authors in mind that the data is “logged”? What means “stabilizing data”? I assume the data is given, what has to be stabilized?

Reviewer 2 Report

The paper identifies and establishes the relationships between environmental regulation, green technologies' innovation, and government subsidies in China. The article is very well-structured and is suitable for the journal.

Authors should pay attention to references and use the journal's standard. In this article version, the references are very confused.

However, small details should be improved:

  • Lines 41-43: Please explain better this sentence, namely the “...achieving carbon peak and being carbon neutral...”
  • Lines 239-250: Please explain better the dataset of 2003-2017 and all data used in this paper from 2004-2018.
  • Lines 314-339: Please explain the need for logarithm transformation in some control variables and others not.
  • Lines 345-347: Although the en_ru values vary a lot, the standard deviation is very low and the average is also low, denoting certainly the presence of outliers. Please rewrite the sentence.
  • Lines 382-387: References are missing
  • Fig 2: Please change the figures corresponding to (a) and (b) in order to be coherent with the text of lines 355-362.
  • Line 48: Please replace “(Porter and Linde, 1995).” by “(Porter and van der Linde, 1995).”
  • Line 104: Please replace “(Jaffe, 1997, Van and Mohnen, 2017).” by “(Jaffe and Palmer, 1997, van Leeuwen and Mohnen, 2017).”
  • Lines 106, 121, 149, 185-186, 197, 212: the reference Chen et al., 2019 and Chen, 2019 are missing
  • Line 108: Please replace “...Li et al., 2020...” by “...Li and Wang, 2020...”
  • Line 211: Please replace “...the square of\ en_ru is...” by “...the square of en_ru is...”
  • Line 214: Please replace “...ß2 – ß1 and δi...” by “...ß1, ß2 and δi...”
  • Line 228: Please replace “...Kashdan (2007),...” by “...Kashdan and Breen (2007),...”
  • Line 290: Please replace “...Heranandez et al...” by “...Hernandez-Sancho et al...”
  • Line 295: Please replace “...(SO2)...” by “...(SO2)...”
  • Line 474: Please replace “...(González, 2018).” by “...(Guisado-González et al., 2018).”
  • Line 553: Please replace “...Jíménez, 2001),...” by “... De Burgos Jiménez and Céspedes Lorente, 2001),...”
  • Reference 12: Please replace “...Heranandez...” by “...Hernandez...”
  • Reference 21: Please replace the reference by “Kashdan, T.B., Breen, W.E. 2007. Materialism and Diminished Well–Being: Experiential Avoidance as a Mediating Mechanism. Journal of Social and Clinical Psychology, 26 (5). https://doi.org/10.1521/jscp.2007.26.5.521
  • Reference 44: Please replace the reference by “Porter, M.E., van der Linde, C. 1995. Toward a New Conception of the Environment-Competitiveness Relationship. Journal of Economic Perspectives, 9 (4): 97-118. DOI: 10.1257/jep.9.4.97
  • Reference 47: Please replace the reference by “Im, K.S., Pesaran, M.H., Shin, Y. 2003. Testing for unit roots in heterogeneous panels. Journal of Econometrics, 115 (1): 53-74. https://doi.org/10.1016/S0304-4076(03)00092-7
  • Reference 53: Please replace the reference by “Wang, H., Davidson III, W.N., Wang, X. 2010. The Sarbanes-Oxley Act and CEO tenure, turnover, and risk aversion. The Quarterly Review of Economics and Finance, 50 (3): 367-376. https://doi.org/10.1016/j.qref.2010.03.005
  • References 56 and 64 are not mentioned in the article.
  • Some references should be completed/corrected (e.g. 14, 22, 24, 25, 27, 28, 30, 31, 36, 38, 43, 52, 55, 57, etc.)

Reviewer 3 Report

General remarks

The manuscript intends to analyze the impact of environmental regulation on green technology innovation and the role of government subsidies in the form of that impact. For this, a panel of data from some provinces of China between 2003 and 2017 is considered. Through a panel data methodology, fixed effects and 2SLS models were estimated, being concluded that the relationship between those variables exhibits spatial heterogeneity.

Specific remarks

For me, this is a standard manuscript that, in scientific terms, meets the requirements, in terms of content and form, for publication. In fact, reading it does not offer any difficulties or great doubts. Still, I would like to draw attention to two aspects of a methodological nature:

  • Given that there is spatial heterogeneity between regions, its existence within each region makes the use of spatial econometric models with panel data recommendable. See, among others, Anselin, L., Le Gallo, J., & Jayet, H. (2008). Spatial panel econometrics. In The Econometrics of Panel Data (pp. 625-660). Springer, Berlin, Heidelberg.

  • The authors should bear in mind that, in models with one of the explanatory variables, say x, in level is also considered as being so in squared terms, x2, the interpretation of the coefficients of these two variables can be ‘tricky’, since the assumption “all else equal” may not be verified when, for example x varies and obviously x2 varies as well. The same argument applies when, besides x, another explanatory variable is x multiplied by another (explanatory) variable.

As minor remarks,

  • Please remove the \ in “[…] of\ en_ru […]” (page 5: 211);

Can you please provide the value of the Hausman test statistic? (page 10: 386).

Reviewer 4 Report

This study examines how environmental regulation affects green technological innovation and investigates the role of government subsidies on this process. While I see this paper intriguing and meaningful, there are some points where the authors further elaborate and develop. Let me discuss those.

  1. Terms

Terms should be made consistent. For example, green innovation and green technological innovation are interchangeably used. I don’t think these terms are so distant in its meaning, but the authors may want to clarify which is the term this paper deals with. Also, in the hypotheses, rather than environmental regulation, environmental regulation intensity could be clearly stated throughout the paper.

  1. Measures

My main concern on this paper is variable measurement. Green (technological) innovation was measured as new product sales revenue divided by total energy consumption. I’m not sure if this measure can completely capture the concept of green innovation. Sales of new products can be an output of innovation, but this cannot tell whether the product is made by new technology or is technologically driven. The revenues themselves indicate (and thus are largely used for) the firm’s performance. The denominator of this ratio measure doesn’t make sense to me. How can we understand the ratio and its scale? When the ratio increases, it occurs due to an increase in revenues or a decrease in energy consumption? The authors may want to elaborate how this measure directly, explicitly, and clearly capture “green technological innovation”. Also, the measure of governmental regulation is misleading. It was measured with the level of pollutants, which reveals how the given firm generates pollutants regardless of governmental regulations. As prior literature showed the way to capture the governmental regulations, the authors may want to measure governmental regulations in terms of how far the given firm’s pollution is from the regulation standards.

As the main variables cannot precisely capture the concepts, the results and their interpretations cannot support the hypotheses. The authors may want to re-develop the hypotheses or re-measure the focal variables.

  1. Testing U-shaped relationships

To test a U-shaped relationship, please cite Haans et al., (2016) SMJ paper. The paper provides guidelines how such U-shaped relationships can be tested and how moderation effects on such a non-linear relationship can be examined.

Haans, R. F., Pieters, C., & He, Z. L. (2016). Thinking about U: Theorizing and testing U‐and inverted U‐shaped relationships in strategy research. Strategic management journal37(7), 1177-1195.

Hope these comments help to develop the paper.

Round 2

Reviewer 1 Report

The paper is still suffering from the fact, that the used indicators partly do not measure what they shall measure. Accordingly, the paper does not make sense. 

Reviewer 4 Report

Thank you for the revision. This revision shows much improvement. However, I'm still not sure about the green innovation measure. I understand how you got the measure, but it is really hard to interpret when you get the results. if the measure increases, it comes from an increase in sales or an decrease in energy consumption? What if the total consumption has been drastically decreased while the sales have been little improved or even slightly decreased? I imagine the measure still shows an improved "green innovation." In this case, you may want to explain the decrease in energy consumption with other factors, like this COVID situation or some improvement of energy consumption. But the measure itself cannot explain how the reduced energy consumption indicates green innovation unless it is driven by technological development for energy efficiency. 

In the references you gave me, I found that only Guo et al.(2018) used the ratio between energy consumption and revenue. But they focused on energy reduction as the main indicator for green innovation, which is opposite to yours. As you mentioned, the reciprocals of your measure may be technically equivalent to Guo et al. (2018)'s measure, but those are totally different in terms of their meanings. To avoid any confusion on the measure, I recommend that all the other citations which didn't compare the energy consumption and the sales of new products should be removed and explicitly note this measure is followed by Guo et al. (2018)'s. And the explanation on the measure should be followed by Guo et al. (2018). Otherwise, readers may have a lot of confusion on the measurement.

Hope this comment help.
